# A potent KRAS macromolecule degrader specifically targeting tumours with mutant KRAS

Nicolas Bery [1,2], Ami Miller[1,3] & Terry Rabbitts [1,3 ✉]

Tumour-associated KRAS mutations are the most prevalent in the three RAS-family isoforms and involve many different amino-acids. Therefore, molecules able to interfere with mutant KRAS protein are potentially important for wide-ranging tumour therapy. We describe the engineering of two RAS degraders based on protein macromolecules (macrodrugs) fused to specific E3 ligases. A KRAS-specific DARPin fused to the VHL E3 ligase is compared to a pan-RAS intracellular single domain antibody (iDAb) fused to the UBOX domain of the CHIP E3 ligase. We demonstrate that while the KRAS-specific DARPin degrader induces specific proteolysis of both mutant and wild type KRAS, it only inhibits proliferation of cancer cells expressing mutant KRAS in vitro and in vivo. Pan-RAS protein degradation, however, affects proliferation irrespective of the RAS mutation. These data show that specific KRAS degradation is an important therapeutic strategy to affect tumours expressing any of the range of KRAS mutations.

[1] Weatherall Institute of Molecular Medicine, MRC Molecular Haematology Unit, University of Oxford, John Radcliffe Hospital, Oxford OX3 9DS, UK. [2] Present address: Cancer Research Centre of Toulouse, INSERM - Université Toulouse III Paul Sabatier - CNRS, 2 avenue Hubert Curien, Toulouse 31037, France. [3] Present address: Institute of Cancer Research, Division of Cancer Therapeutics, 15 Cotswold Road, Sutton, London SM2 5NG, UK. ✉email: terry.rabbitts@icr.ac.uk

M utations of the *KRAS* oncogene represent more than 85% of all RAS family mutations[1] and individual mutations occur at various codons giving rise to many forms of mutant KRAS protein[2]. Recently, several macromolecules[3–7] and compounds[8–13] have been developed that influence the function of RAS family members. Nevertheless, only the G12C mutation of KRAS has been specifically targeted by small molecules by virtue of covalent binding of the compounds to the mutant cysteine[14–21]. Several small molecules are now in clinical trials[22]. However, only a small portion of mutant KRAS tumours expresses a KRAS$^{G12C}$ protein (around 12%, Cosmic database v91, https://cancer.sanger.ac.uk/cosmic) and can be targeted by these inhibitors. Furthermore, upon treatment with these inhibitors, a rapid adaptation has been recently described in the KRAS$^{G12C}$ tumour population such that some cells become drug-insensitive[23]. Therefore, new strategies are needed to specifically target the larger number of other tumours expressing mutant KRAS. Reagents that could potentially be applied to this objective are designed ankyrin repeat proteins (DARPins) K13 and K19 that interfere specifically with KRAS[3]. While these DARPins do not discriminate mutant from wild-type KRAS (KRAS$^{WT}$), they do not bind to NRAS and HRAS so that any phenotype engendered using these DARPins within cells would spare the expression, and function, of these two family members[3].

Previous studies involving expression of a pan-RAS-binding intracellular single domain antibody (iDAb, hereafter called iDAb RAS) in human cell line xenografts demonstrated that tumour growth was inhibited for the duration of expressing the intracellular antibody fragment and resumed when the antibody was removed[24]. A potential complementary direction in this context could be the addition of warheads to these macromolecules[25], such as single chain Fragment variable (scFv) fused to proteasome targeting sequences[26] or E3 ligases engineered on intracellular single domains, called macrodrug degraders, that have been shown to invoke proteolysis of targets[27–32]. Macromolecule degraders induce the depletion of their target via the ubiquitin-proteasome system. They consist of a binder targeting a protein of interest (e.g. intracellular single domain), a linker and an E3 ligase domain. A similar protein target degradation strategy has been developed in which small molecules that bind proteins are linked to E3 ligase-binding ligands called proteolysis targeting chimeras (PROTACs) or small molecule degraders[33,34].

The main advantage of the proteolysis strategy is that only a binder is required, and the binder does not need to inhibit the function of the protein. Indeed, unlike classical protein–protein inhibitors or other occupancy-driven inhibitors, the degraders rely on an event-driven mode of action and are consequently often more potent than the parental entity[35–39]. Most of the current degraders target bromodomains or kinase families[40–43] and only a few target "undruggable" proteins such as transcription factors[44]. The only PROTACs thus far applied to RAS are compounds that bind KRAS$^{G12C}$ but these only degrade exogenous GFP–KRAS$^{G12C}$ fusion protein and do not target endogenous KRAS$^{G12C}$[45]. In addition, no macromolecule or small molecule-based degrader has been shown to be specific to KRAS in the RAS family of oncogenic targets.

We report here the engineering of the KRAS-specific DARPin K19 (herein referred to as DP KRAS)[3] into a KRAS-specific degrader and compare the efficacy and tumour-specificity with an engineered pan-RAS degrader, made from the previously described pan-RAS intracellular single domain antibody[7,46]. We show that the KRAS degrader efficiently induces endogenous KRAS degradation in vitro and in vivo and specifically inhibits mutant KRAS tumours without affecting cells with only KRAS$^{WT}$, whereas the pan-RAS degraders inhibit all type of cells, regardless of the RAS isoform mutation. Therefore, we exploit

this KRAS-specific macrodrug to demonstrate that KRAS ablation can be an attractive way to target any mutant KRAS-expressing tumour.

## Results

**Engineering KRAS-specific and pan-RAS degraders**. The addition of warheads to intracellular antibodies and other macromolecular reagents is an implied strategy to increase efficacy, for example via the transfer of target proteins to the proteasome for degradation. Intracellular single domains have been functionalised with E3 ligase domains, such as the UBOX domain of the carboxyl terminus of Hsc70-interacting protein (CHIP) ligase[31], the Von Hippel–Lindau (VHL)[29,32] or FBOX[27,28] but there are no rules about which E3 ligase is applicable in specific cases nor how the protein should be engineered (N or C terminal fusion with the E3 ligase). We have tested both the UBOX domain and VHL E3 ligase fused to the specific DP KRAS[3] or the iDAb RAS[7]. Controls comprised a mutant DARPin where the RAS-binding tryptophan repeats are mutated into glycine and alanine residues[3] (herein DP Ctl) or a non-relevant iDAb (herein iDAb Ctl)[47]. All the proteins were engineered with either N or C-terminal fusions with each E3 ligase. Accordingly, DP Ctl does not bind to KRAS (mutant and WT) as shown by Bioluminescence resonance energy transfer (BRET) donor saturation assays (Supplementary Fig. 1a) and by co-immunoprecipitation (Supplementary Fig. 1b). Furthermore, like the negative control DARPin E3.5[4], the DP Ctl mutant does not inhibit mutant KRAS/CRAF$^{FL}$ interaction (Supplementary Fig. 1c) or mutant KRAS dimerisation in BRET competition assays (Supplementary Fig. 1d).

HCT116 cells (which express KRAS$^{G13D}$) were transiently transfected with these UBOX/single domain constructs and K/N/HRAS protein levels were monitored by Western blot. The iDAb RAS-UBOX and UBOX-iDAb RAS constructs both induced a decrease of KRAS, NRAS and HRAS protein levels, and the iDAb RAS-UBOX showing greater degradation (Fig. 1a, b). Conversely, we could detect little RAS turnover by the C-terminal VHL fusion with the iDAb RAS (i.e. iDAb RAS-VHL), although some degradation was observed with the N-terminal VHL fusion (i.e. VHL-iDAb RAS) (Fig. 1c, d). Similarly, UBOX fusions with the DP KRAS showed little effect on RAS protein levels (Fig. 1a, b). However, when we engineered a VHL fusion at the N-terminus of DP KRAS (VHL-DP KRAS), we observed a substantial decrease of KRAS protein in the transfected HCT116 cells (Fig. 1c, d). VHL-DP KRAS fusion diminished KRAS protein level much more than DP KRAS-VHL fusion (Fig. 1d). A key observation is that, unlike iDAb RAS-UBOX, VHL-DP KRAS only depleted KRAS and not NRAS or HRAS (Fig. 1c, d) due to the KRAS-specific binding property of this DARPin[3]. These degradation effects were proteasome dependent as epoxomicin treatment (a proteasome inhibitor[48]) impeded RAS degradation for the iDAb-UBOX and the VHL-DP KRAS (Fig. 1e, f). Therefore, the functionalisation of iDAb RAS with UBOX domain (hereafter called pan-RAS degrader) and DP KRAS with VHL (hereafter named KRAS degrader) promotes RAS and KRAS degradation, respectively (Fig. 1g, the sequences in Supplementary Figs. 1e and 2–4).

**KRAS degrader specifically depletes KRAS in cancer cells**. We performed a comparison of the KRAS-specificity of the KRAS degrader with the pan-RAS degrader in a stably transduced H358 (lung, KRAS$^{G12C}$) cancer cell line. The engineered proteins were expressed using a Tet-On inducible system (i.e. degrader expression is induced after doxycycline treatment) in cells transduced with lentiviral vectors. We first characterised the degrader properties in these H358 cells with an increasing dose of doxycycline induction, demonstrating a depletion of only KRAS

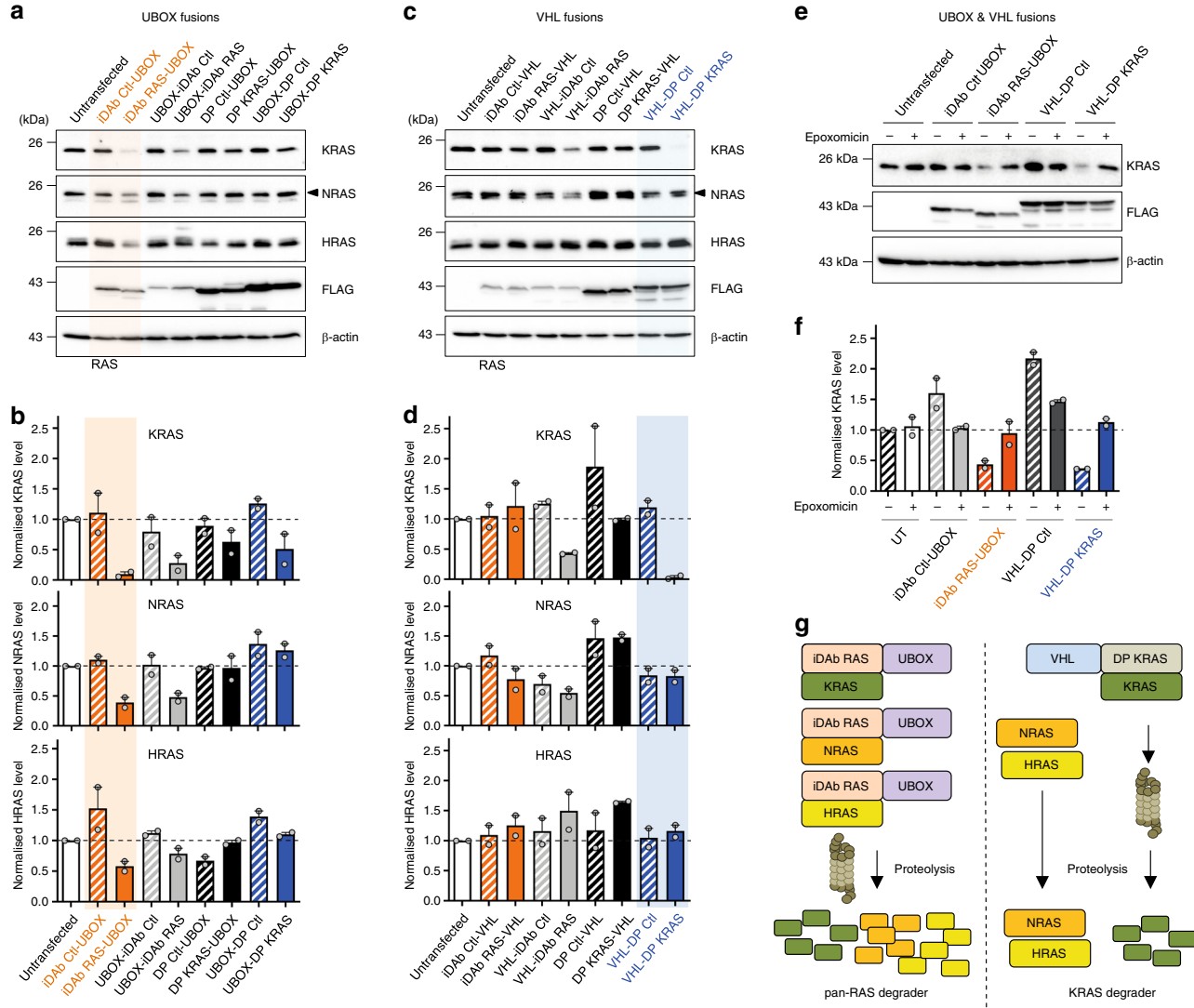

**Fig. 1 Engineering KRAS and pan-RAS-targeted protein degradation with single domains. a** Effect of the fusion single domain-UBOX on RAS proteolysis assessed by Western blots. The UBOX domain was fused either on the N- or C-terminal end of the pan-RAS iDAb (iDAb RAS) or iDAb Control (Ctl), the KRAS-specific DARPin K19 (DP KRAS) or DARPin Control (DP Ctl). Twenty-four hours after transfection of the plasmids in HCT116 cells, a Western blot was performed to determine RAS protein levels and the expression from each construct with a FLAG antibody. β-actin is the loading control. The black arrowhead shows the specific band corresponding to NRAS protein in the observed doublet. **b** The histograms display the quantification of each RAS isoform from two independent biological repeats (grey dots) normalised to untransfected cells. **c** Effect of the fusion single domain-VHL on RAS proteolysis assessed by Western blots as in **a**. **d** Quantification of two independent biological repeats (grey dots) normalised to the untransfected cells as in **b**. In **a–d**, the best pan-RAS and KRAS degraders have been highlighted in orange and blue, respectively. **e** HCT116 transfected cells with the indicated constructs were treated with DMSO (−) or 0.8 μM of the proteasome inhibitor epoxomicin (+) for 18 h. KRAS degradation was evaluated by Western blots. **f** KRAS degradation quantification of two independent biological repeats normalised to the untransfected cells. Experiments in **a**, **c e** were performed twice. Error bars in **b**, **d** and **f** are mean ± SEM from two independent biological repeats. **g** Scheme recapitulating the effect of the two RAS degraders: the pan-RAS degrader (iDAb RAS-UBOX) degrades all RAS isoforms through the proteasome machinery while the KRAS degrader (VHL-DP KRAS) only induces the degradation of KRAS without affecting NRAS or HRAS. **a–f** Source data are provided as a Source Data file.

with the KRAS degrader while the pan-RAS degrader knocked down all three K/N/HRAS proteins, in a dose dependent manner (Fig. 2a, b). No effect was observed with the control degraders (Fig. 2a, b). These results confirmed the specificity of degradation observed in the transient transfection experiments shown in Fig. 1. We analysed the kinetics of RAS protein degradation following doxycycline induction of either the pan-RAS degrader or the KRAS degrader and the resultant effects of RAS-dependent downstream signalling. The synthesis of degrader proteins (detected using Western blot with anti-FLAG antibody) could be observed in as little as 2 h after doxycycline treatment (Fig. 2c, d) and the loss of K, N and HRAS followed with similar profiles

from 2 h when the pan-RAS degrader was expressed (Fig. 2c). However, only the KRAS level was reduced when the KRAS degrader was expressed and the KRAS reduction was concomitant with the degrader expression (Fig. 2d). As expected, we also observed loss of phosphorylation of AKT, MEK and ERK in parallel with either degrader synthesis (Fig. 2e, f). RAS degradation was confirmed to be proteasome-dependent by treating cells with the proteasome inhibitor epoxomicin (Supplementary Fig. 5a). In addition, consistent with the potent inhibition of MAPK pathway activity, both pan-RAS and KRAS degraders reduced the expression of the MAPK pathway downstream transcript *DUSP6* (Supplementary Fig. 5b). Our data demonstrate

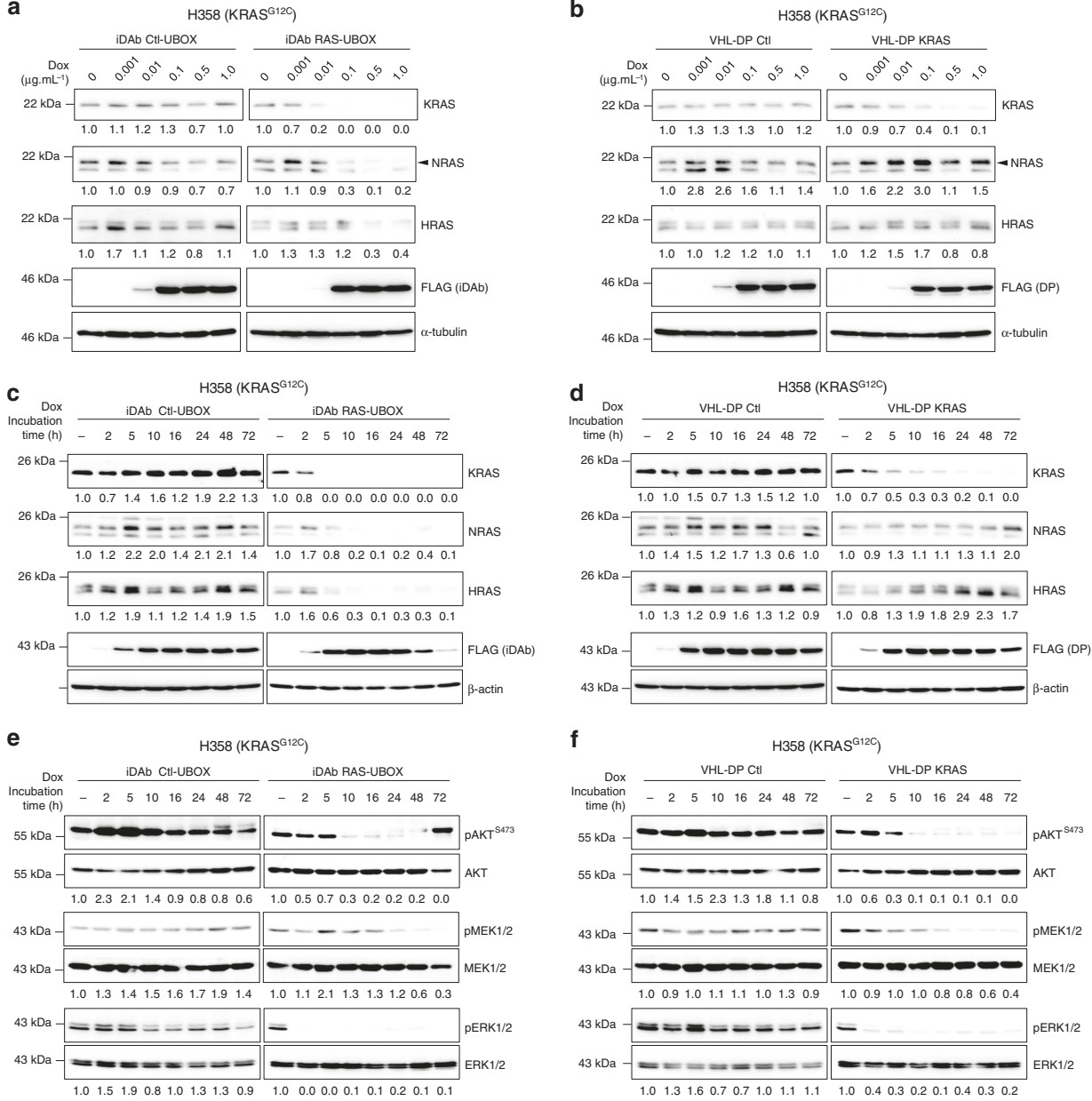

**Fig. 2 Characterisation of pan-RAS and KRAS degraders in H358 cancer cells.** Western blot of RAS and effectors protein levels following induction of the macrodrug degraders for dose response and time course analysis. **a**, **b** Dose response experiment with the indicated doxycycline (dox) concentrations incubated for 18 h in H358 cells expressing either iDAb Ctl-UBOX or iDAb RAS-UBOX (**a**), VHL-DP Ctl or VHL-DP KRAS (**b**). KRAS, NRAS and HRAS protein levels were assessed by Western blot. FLAG antibody shows the expression of the constructs and α-tubulin is the loading control. The black arrowhead shows the specific band corresponding to NRAS protein in the observed doublet. **c**, **d** Time response experiment of H358 cells expressing the same degraders as in **a**, **b**. Effect of the pan-RAS (**c**) and KRAS degraders (**d**) on KRAS, NRAS and HRAS protein levels was determined using β-actin as the loading control. **a**–**d** α-tubulin/β-actin-normalised KRAS, NRAS and HRAS abundance values are reported beneath individual lanes. **e**, **f** Effect of the pan-RAS (**e**) and KRAS degraders (**f**) on RAS dependent pathways was assessed and the β-actin loading control is the same as in panels **c**, **d** as the panels were split for clarity purpose. Quantifications of pAKT$^{S473}$/AKT, pMEK/MEK and pERK/ERK signals normalised to the no dox (−) condition are shown underneath individual lanes. Each experiment was performed twice. **a**–**f** Source data are provided as a Source Data file.

the KRAS-specific degrader causes rapid depletion of KRAS coupled to an inhibition of RAS downstream signalling and sustained degradation of KRAS over the doxycycline treatment period (as long as 72 h). The pan-RAS degrader shows similar properties.

We also assessed the specificity of degradation of both degraders in several stably-transduced cancer cell lines in addition to H358, namely MIA PaCa2 (pancreas, KRAS$^{G12C}$), A549 (lung,

KRAS$^{G12S}$), H1299 (lung, NRAS$^{Q61K}$), HT1080 (fibrosarcoma, NRAS$^{Q61K}$), T24 (bladder, HRAS$^{G12V}$), HCC827 (lung, RAS$^{WT}$ but EGFR mutated) and an untransformed cell line: MRC5 (non-transformed lung fibroblast, RAS$^{WT}$). The engineered proteins were again expressed using a Tet-On inducible system in cells transduced with lentiviral vectors. The degrader expression was detected in Western blots using anti-FLAG tag antibody on induction by doxycycline (Fig. 3) causing efficient knockdown of

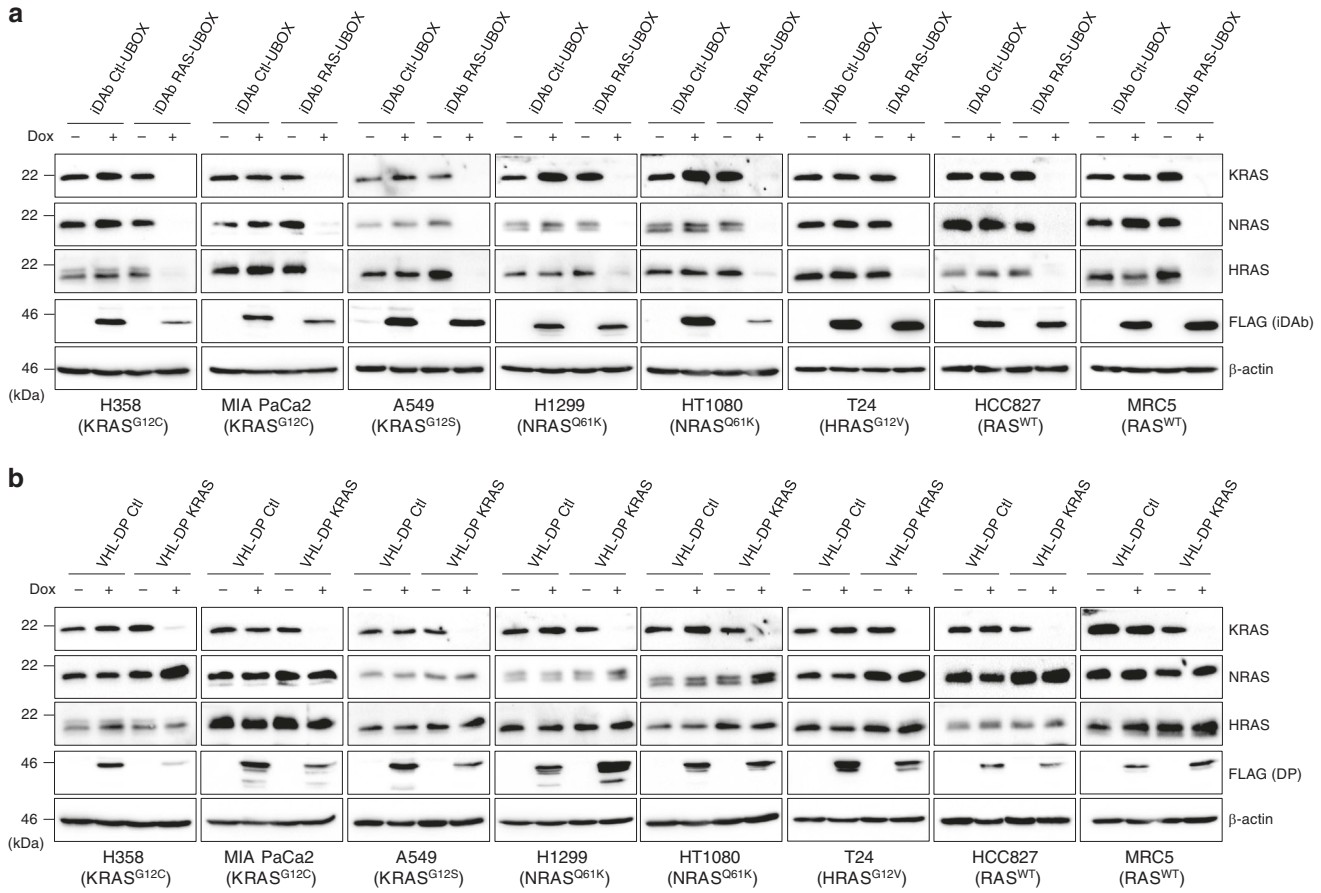

**Fig. 3 Efficacy of pan-RAS and KRAS degraders in cancer cells.** Western blot of RAS family protein levels following induction of the macrodrug degraders. **a**, **b** Efficacy of the pan-RAS (**a**) and KRAS degraders (**b**) to deplete their target(s) in various human cell lines with the indicated variety of RAS mutations. KRAS, NRAS and HRAS protein levels were judged by Western blots after 72 h of doxycycline treatment of the cells at 0.2 μg.mL$^{-1}$ (+) or not induced (−). The lines used were mutant KRAS cell lines H358, MIA PaCa2 and A549, mutant NRAS H1299, HT1080 and mutant HRAS T24 cell lines and RAS$^{WT}$ HCC827 and MRC5 cells. Expression of the degraders is shown with the FLAG antibody. β-actin is the loading control. Each experiment in **a**, **b** was performed twice. **a**, **b** Source data are provided as a Source Data file.

K, N and HRAS in cells expressing the pan-RAS degrader (Fig. 3a) or degradation of KRAS only when the KRAS degrader was expressed (Fig. 3b). This occurred in all the cell lines tested.

**KRAS degrader inhibits RAS signalling of mutant KRAS cells.** The kinetics of RAS degradation in H358 cells was examined to determine the initiation and duration of effects on RAS signalling (Fig. 2). We further evaluated the consequences of pan-RAS or KRAS-only degradation on RAS downstream signalling pathways of the panel of cell lines using Western blots 72 h after induction with doxycycline. The pan-RAS degrader induced loss of RAS proteins which resulted in the inhibition of RAS signalling (either PI3K and/or MAPK pathways). This was determined by reduction in phosphorylation of AKT, MEK and/or ERK, in all the cell lines, although not all proteins in the MAPK and PI3K pathways were similarly affected (Figs. 4 and 5i). Loss of RAS signalling was also observed in two lines without *RAS* gene mutations (Fig. 4g, h), whereas the control iDAb Ctl-UBOX had no effect (Fig. 4a–h). Conversely, the KRAS degrader only inhibited AKT, ERK and MEK phosphorylation in those cells with mutant KRAS, namely H358, MIA PaCa2 and to a lesser extent in A549 (Fig. 5a–c, i). Indeed, it had no consequence on cells with mutation of NRAS (H1299 and HT1080) or HRAS (T24) or without mutant RAS (HCC827 and MRC5) (Fig. 5d–h). The VHL-DP Ctl had no effect (Fig. 5). These data are quantified in Fig. 5i. It has been previously reported that the conversion of a parental binder into a degrader

could change its specificity[49–51], so we added the parental binders as controls in our study. Expression of the pan-RAS iDAb fused to GFP[2] resulted in reduced RAS levels in H358, A549 and T24 (Supplementary Fig. 6a–h) that could be due to the level of expression of the iDAb RAS, while the KRAS-specific DARPin fused to GFP[2] had no effect on the levels of RAS proteins (Supplementary Fig. 7a–h). The iDAb RAS-GFP[2] had a similar inhibitory output on RAS downstream pathways than its degrader version (Supplementary Figs. 6a–h and 7i for quantification). The KRAS-specific DARPin fused to GFP[2] also decreased RAS signalling in mutant KRAS cells (H358, MIA PaCa2 and A549 in Supplementary Fig. 7a–c, i), whereas it altered RAS signalling differently in the other cell lines (Supplementary Fig. 7d–i). Indeed, it augmented MEK and ERK phosphorylation, while it decreased or had no effect on pAKT levels in H1299, HT1080 and T24 cells, which have mutant NRAS or HRAS (Supplementary Fig. 7d–f, i) and diminished pERK and pAKT in HCC827 and MRC5 cell lines lacking RAS mutation (Supplementary Fig. 7g–i). The KRAS-specific DARPin interacts with KRAS-GTP and KRAS-GDP and may be a GAP inhibitor[3], therefore, the increase of pMEK/pERK signals might be attributed to its GAP inhibitory mechanism on KRAS$^{WT}$.

**KRAS degrader inhibits mutant KRAS cells proliferation.** Our data showed that the KRAS-specific DARPin K19 engineered into a degrader has focused specificity towards mutant KRAS cells

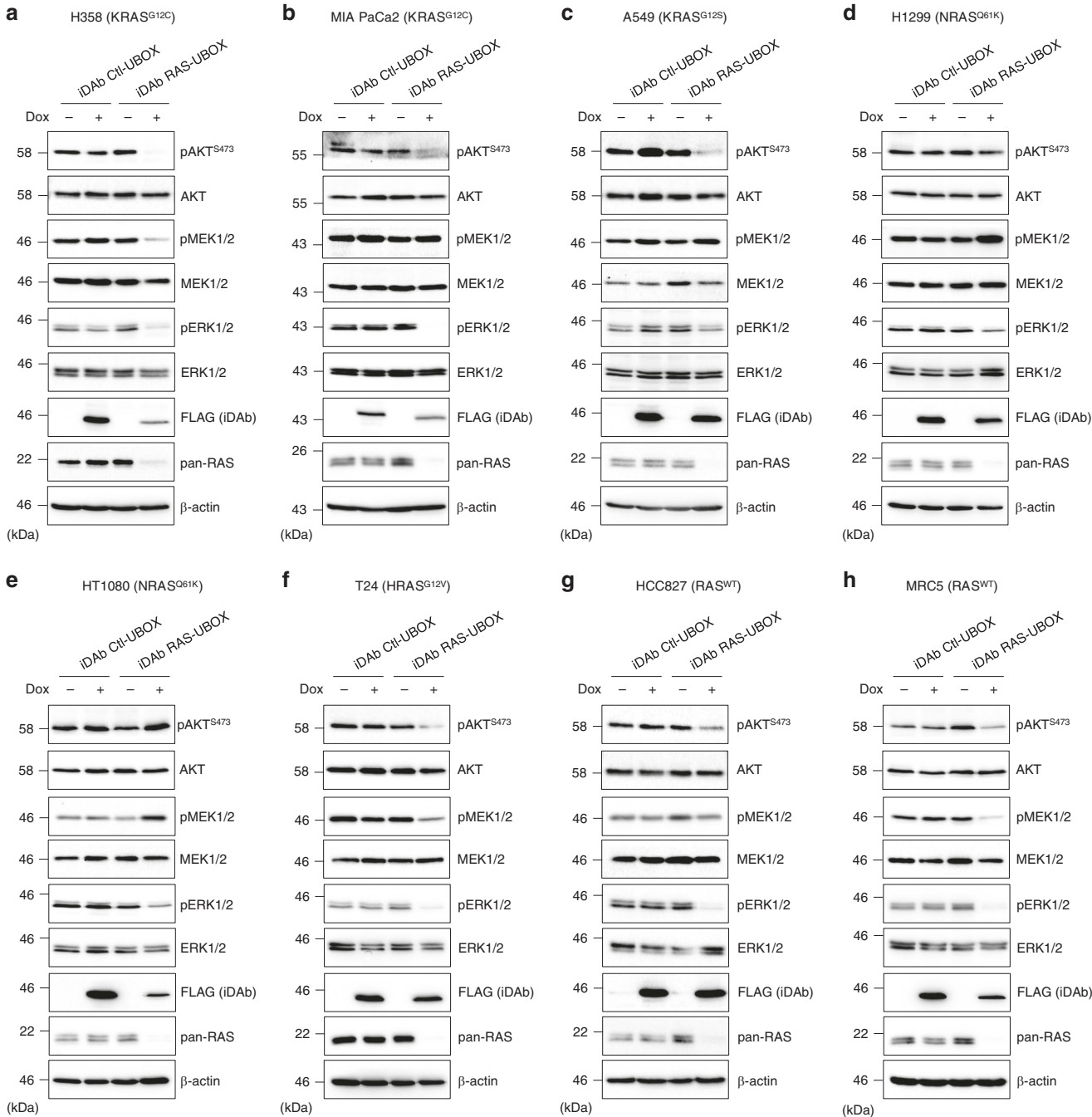

**Fig. 4 The pan-RAS degrader inhibits RAS signalling pathways in all cell lines.** Effect of the pan-RAS degrader on RAS downstream signalling pathways of various cell lines was examined by Western blot analysis: **a** H358 (KRAS$^{G12C}$), **b** MIA PaCa-2 (KRAS$^{G12C}$), **c** A549 (KRAS$^{G12S}$), **d** H1299 (NRAS$^{Q61K}$), **e** HT1080 (NRAS$^{Q61K}$), **f** T24 (HRAS$^{G12V}$), **g** HCC827 (RAS$^{WT}$) and **h** MRC5 (RAS$^{WT}$). All the cell lines stably express dox inducible iDAb RAS-UBOX or its negative control iDAb Ctl-UBOX. FLAG antibody is used to determine iDAbs expression when induced with 0.2 μg.mL$^{-1}$ of doxycycline for 72 h (+) or not induced (−). β-actin is the loading control. Each experiment was performed at least three times. **a–h** Source data are provided as a Source Data file.

compared to the parental DARPin. We assessed the effect of KRAS or pan-RAS degradation on the proliferation of our panel of cell lines with inducible macrodrugs by testing the proliferation in 2D-adherent and 3D spheroid assays. The pan-RAS degrader inhibited proliferation of all the cell lines, in both 2D-adherent or 3D spheroid assays (Fig. 6a–h) including MRC5 which is non-transformed and has wild-type RAS. This was also observed with the parental iDAb RAS-GFP$^2$ in 2D-adherent proliferation assays (Supplementary Fig. 8a–h). The KRAS degrader specifically reduced proliferation of cells with mutant KRAS expression in 2D-adherent and 3D spheroid assays (Fig. 6a–c) but did not

modify the proliferation of cancer cell lines with mutant NRAS or HRAS (Fig. 6d–f) nor of the RAS$^{WT}$ cells HCC827 and MRC5 (Fig. 6g, h). These data concur with the effect of the KRAS degrader described on RAS signalling pathways in Fig. 5. On the contrary, the parental DARPin KRAS-GFP$^2$, like the pan-RAS iDAb-GFP$^2$ and the pan-RAS degrader, decreased the 2D-adherent proliferation of all the stable cell lines (Fig. 6, Supplementary Fig. 8a–h), endorsing the benefit of engineering the DARPin KRAS into a KRAS-specific degrader.

The pan-RAS and the KRAS degraders both inhibited proliferation in H358 cells by inducing programmed cell death

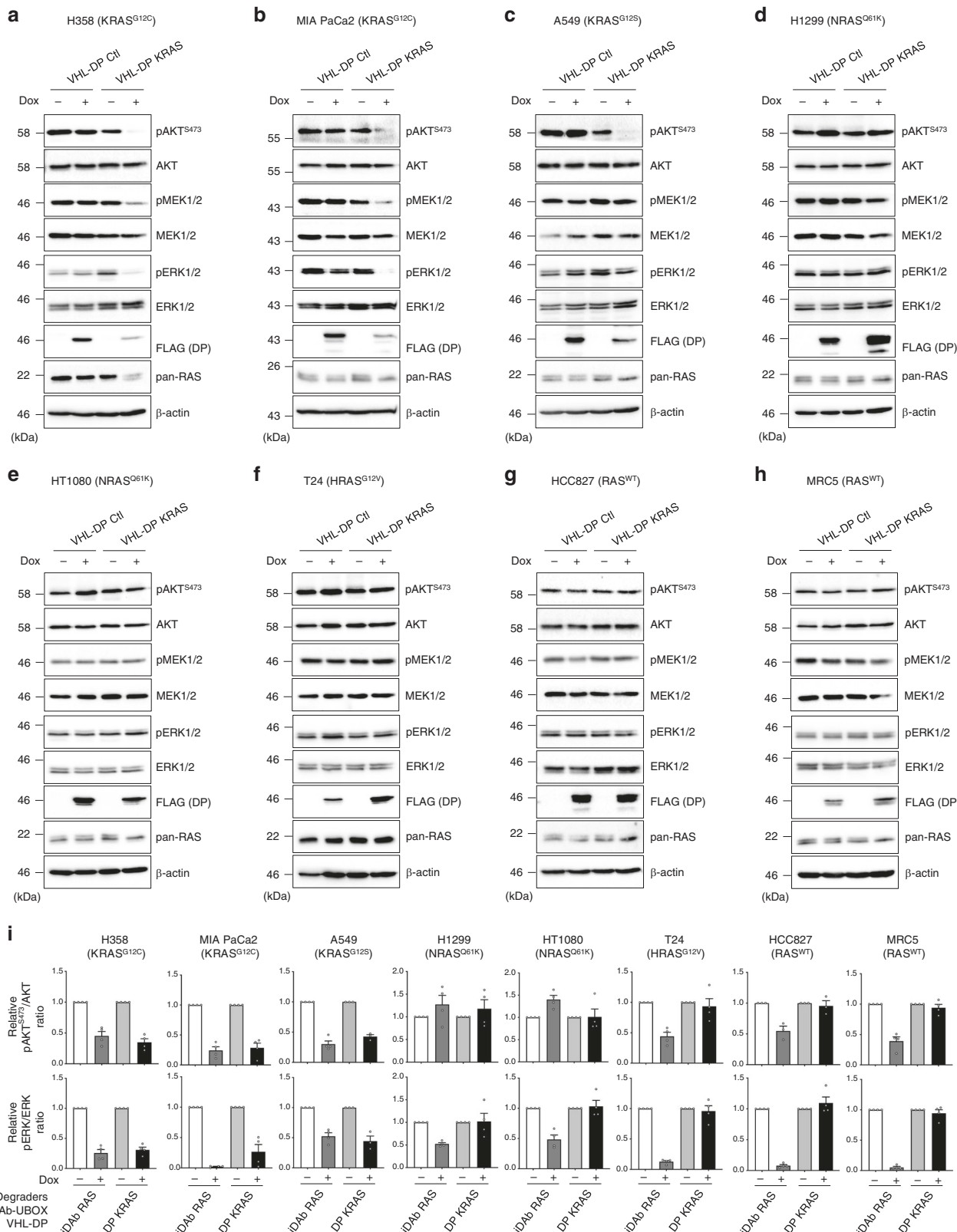

indicated by evidence of the apoptosis markers of cleaved PARP and cleaved caspase 3, starting from 16–24 h after doxycycline addition, with the highest response at 72 h (Fig. 7a). Consequently, we evaluated the induction of apoptosis markers after 72 h of degrader expression in all the stable cell lines. The KRAS degrader and pan-RAS induced cleavage of caspase 3 and PARP in H358 and MIA PaCa2 cell lines but also in A549 cells upon

expression of the pan-RAS degrader only (Fig. 7b). The other cell lines showed little cleaved PARP or caspase 3 compared to the negative controls (Fig. 7b, c). These results suggest that the KRAS degrader blocked proliferation of cells expressing mutant KRAS by inducing apoptosis in KRAS dependent cells (i.e. H358 and MIA PaCa2), while it did not in KRAS independent cells (A549) in the 2D-adherent culture method[52,53].

**Fig. 5 KRAS degrader only inhibits RAS signalling pathways in cancer cell lines expressing mutant KRAS.** The effect of the KRAS degrader on RAS downstream signalling pathways of various cell lines was examined by Western blot analysis: **a** H358 (KRAS$^{G12C}$), **b** MIA PaCa-2 (KRAS$^{G12C}$), **c** A549 (KRAS$^{G12S}$), **d** H1299 (NRAS$^{Q61K}$), **e** HT1080 (NRAS$^{Q61K}$), **f** T24 (HRAS$^{G12V}$), **g** HCC827 (RAS$^{WT}$) and **h** MRC5 (RAS$^{WT}$). All the cell lines stably express dox inducible VHL-DP KRAS or its negative control VHL-DP Ctl. FLAG antibody is used to determine DARPins (DP) expression when induced with 0.2 µg.mL$^{-1}$ of doxycycline for 72 h (+) or not induced (−). β-actin is the loading control. **i** Quantifications of pAKT$^{S473}$/AKT and pERK/ERK signals from Figs. 4 and 5. The signals were normalised to the no dox (−) condition. Each experiment in **a–h** was performed at least three times. Error bars in **i** are mean ± SEM from at least three biological repeats ($n = 3$ for A549 DP KRAS for both quantification and HCC827 for pAKT quantification, all other cell lines $n = 4$). **a–i** Source data are provided as a Source Data file.

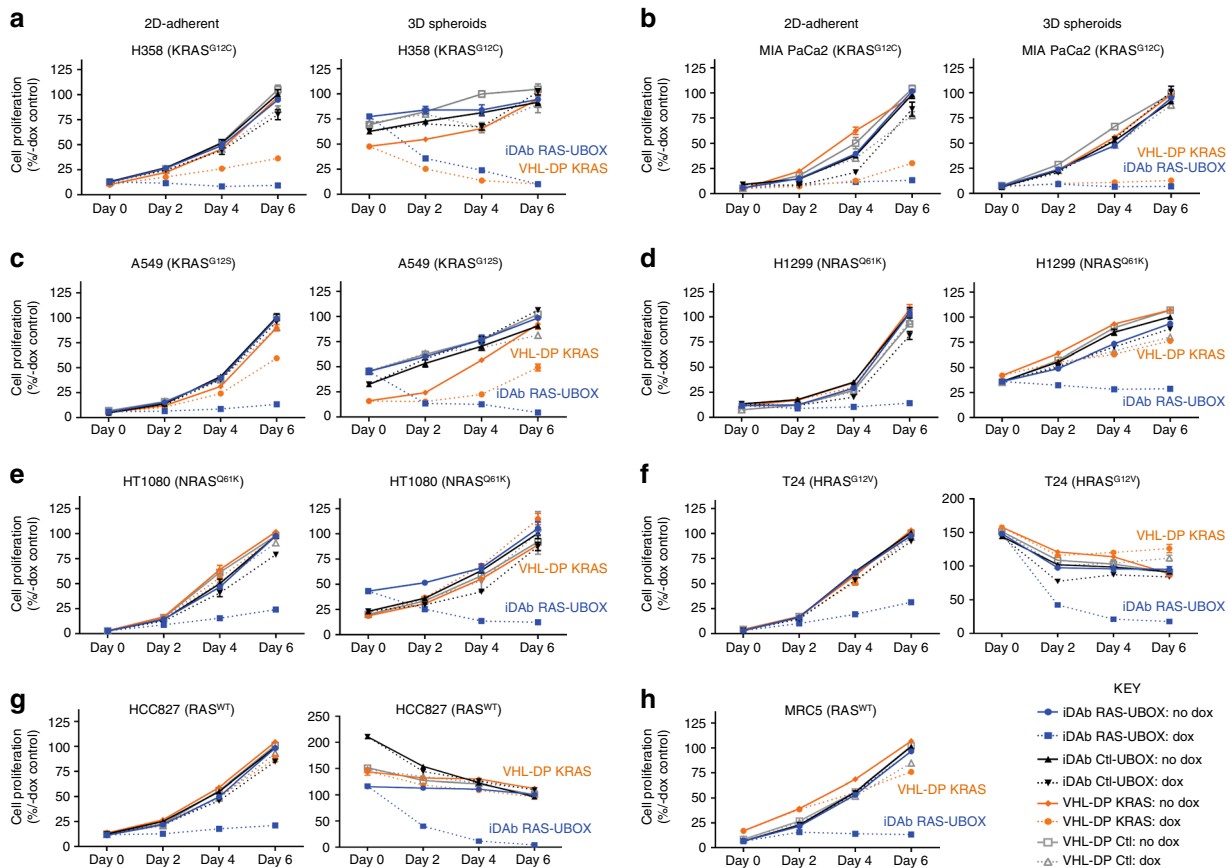

**Fig. 6 The KRAS degrader specifically inhibits the proliferation of cells expressing mutant KRAS.** Assessment of the effect of pan-RAS and KRAS degraders on 2D-adherent and 3D spheroids proliferation of various cell lines. Mutant KRAS lines **a** H358, **b** MIA PaCa2 and **c** A549 stable cell lines. **d**, **e** Mutant NRAS lines: **d** H1299 and **e** HT1080. **f** Mutant HRAS T24 cell line. **g**, **h** Wild-type RAS cell lines: **g** HCC827 and **h** MRC5. Note that MRC5 cells do not grow as spheroids in 3D low attachment plates, since as untransformed cells, they need anchorage to grow. All proliferation assays (2D and 3D) were normalised to the no dox condition for each cell line. The dotted lines represent the dox-treated cells while the plain lines show the no dox conditions. Each experiment in **a–h** was performed at least three times. Error bars in **a–h** are mean ± SD from at least three biological repeats ($n = 3$ in (**a–c**, **g**) and $n = 4$ in (**d–f**, **h**)). **a–h** Source data are provided as a Source Data file.

**KRAS degrader induces regression of mutant KRAS tumours.** We finally determined whether the degraders could be efficient in subcutaneous xenograft mouse models. From our previously established H358 and H1299 stable cell lines, we isolated unique clones of pan-RAS/KRAS degraders that would additionally express a *Firefly Luciferase* (FLuc) to detect the tumour in vivo. These individual clones were characterised in vitro by Western blot and growth curves. Induction of the expression of pan-RAS and KRAS degraders in H358 inhibited RAS downstream signalling pathways (Supplementary Fig. 9a) and strongly impeded the cell growth (Supplementary Fig. 9b). Only the pan-RAS degrader had an inhibitory effect on the RAS signalling pathways and the cell growth of H1299 cells (Supplementary Fig. 9a, c).

while the KRAS degrader had no effect, on either, in H1299 (Supplementary Fig. 9a, c).

The growth of doxycycline-inducible cells in vivo was examined by subcutaneously injecting cells in nude mice to establish xenograft models. While the pan-RAS degrader significantly impeded H1299 tumours burden (Supplementary Fig. 9d), the KRAS degrader caused almost no inhibition (Supplementary Fig. 9e). However, expression of both pan-RAS and KRAS degraders in mutant KRAS H358 xenografts induced regression of tumours after only 3 days of doxycycline treatment with a substantial regression of all tumours after 20 days of treatment (Fig. 8a–d). In order to analyse RAS downstream signalling in H358 xenografts, two mice were treated with

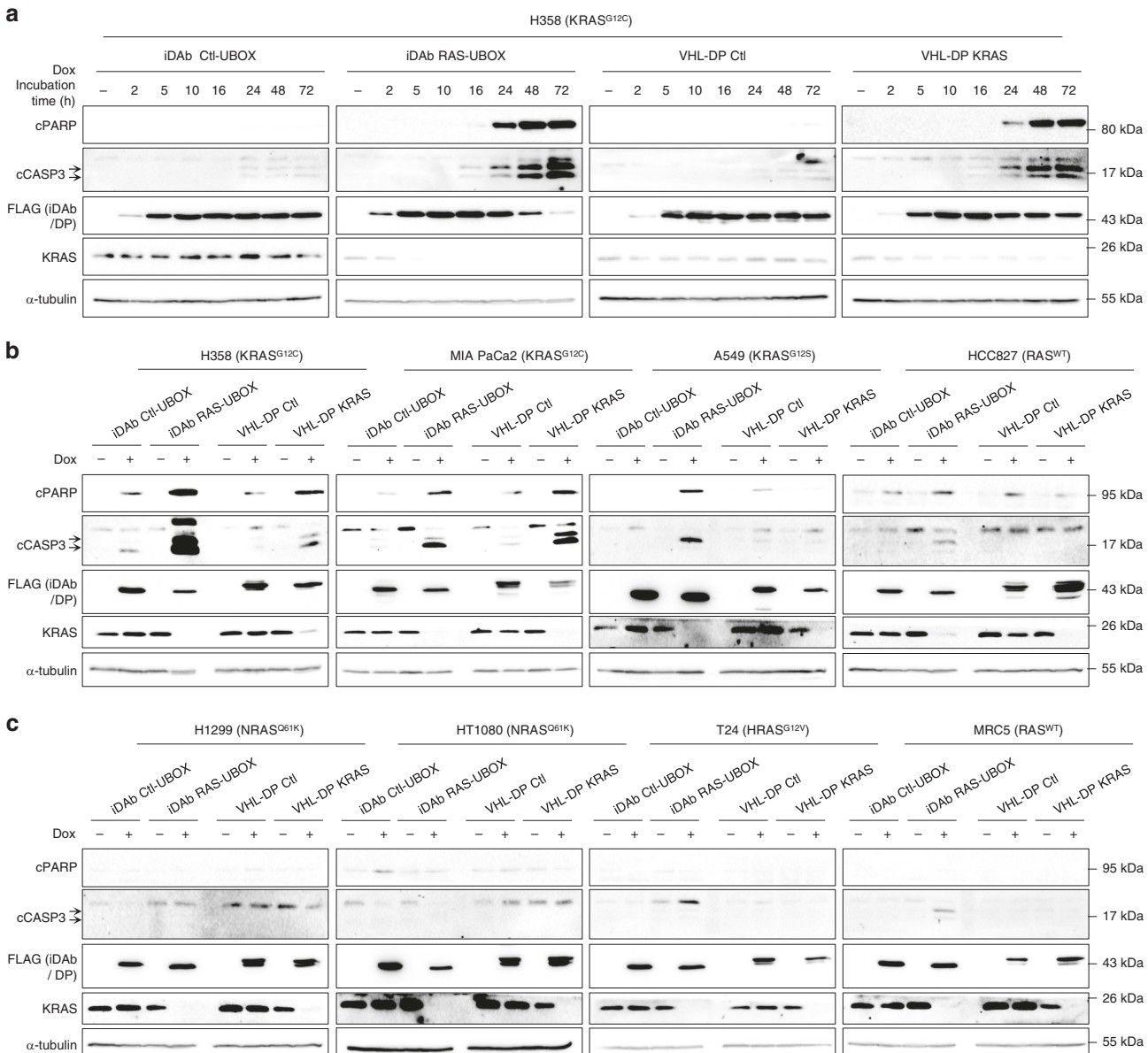

**Fig. 7 KRAS protein depletion by the KRAS degrader leads to apoptosis of mutant KRAS dependent cells. a** Western blot analysis of the apoptosis indicators, viz. cleaved PARP (cPARP) and cleaved caspase 3 (cCASP3) induced after expression of the degraders in H358 cells in a time response experiment. **b**, **c** Western blot analysis of the two apoptosis markers after expression of the degraders in 2D-adherent cultures of all 8 cell lines tested in this study after 72 h of doxycycline treatment at $0.2\,\mu g.mL^{-1}$ (+) or not induced (−). α-tubulin is the loading control. The two arrows indicate the cleaved caspase 3 fragments at 17/19 kDa. Each experiment in (**a–c**) was performed twice. **a–c** Source data are provided as a Source Data file.

doxycycline for 48 h before Western blot analysis of their tumour compared to the non-treated tumours because the doxycycline treated tumours regressed and could not be resected after 20 days of treatment. After 48 h of doxycycline, we observed a large decrease in phosphorylation of MEK and ERK in H358 pan-RAS and KRAS degraders in parallel with degradation of their RAS target(s) (Fig. 8e, f). In contrast, in H1299 tumours, 20 days after treatment, we detected a decrease of phosphorylation of MEK and ERK kinases in tumours expressing the pan-RAS degrader (Supplementary Fig. 9f) and not the KRAS degrader (Supplementary Fig. 9g).

In conclusion, our in vitro and in vivo data show that even though the KRAS degrader depletes both endogenous KRAS^WT and KRAS^MUT, it only inhibits cancer cells expressing mutant KRAS.

## Discussion

Effective cancer therapy based on developing reagents to intracellular targets that comprise families of proteins should ideally incorporate specific targeting of individual family members. The RAS family is an important example in which three isoforms exist, each of which can undergo mutation in various tumour types. KRAS protein is the most often mutated isoform in human cancers, principally resulting from base changes causing single amino acid changes that are spread throughout the protein, but in mutational hotspots[2]. Therefore, targeting mutant KRAS is challenging due to the number of different mutations and the high sequence identity between the three RAS isoforms (more than 80%). However, our recently described KRAS-specific DARPins showed the feasibility to specifically target both wild-type and mutant KRAS by binding on the allosteric lobe of RAS[3].

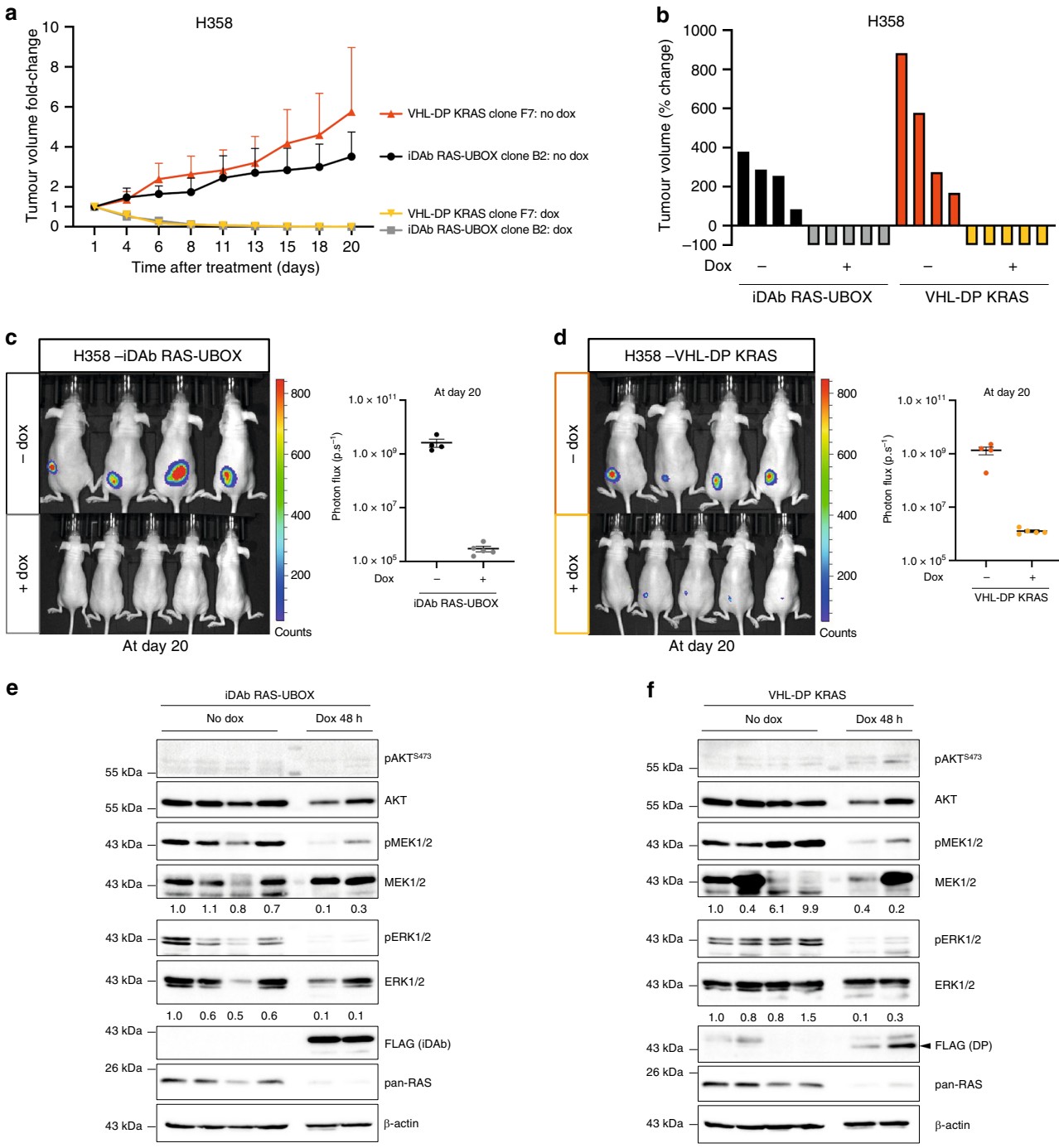

**Fig. 8 The KRAS degrader induces regression of mutant KRAS H358 tumours.** Totally, $4.5 \times 10^6$ H358 cells expressing either FLuc/iDAb RAS-UBOX or FLuc/VHL-DP KRAS were injected subcutaneously into CD-1 nude mice. After tumours reached 3–4 mm diameter, animals were separated into groups of 5 mice and treated or not with doxycycline (dox) in drinking water and food. **a** Tumour volumes were measured using digital calipers and normalised to 1 at the starting day of dox treatment (day 1) and monitored for 20 days for each group ($n = 4$ mice for the no dox condition and $n = 5$ mice for the dox condition, mean ± SD). **b** Waterfall plot representing the percentage of change in tumour volume of individual tumours after 20 days of ± dox treatment. The percentage of change in tumour volume was calculated as follow: $V_{final} - V_{initial} / V_{initial} \times 100$. The colour scale used is the same as in (**a**). **c, d** Tumour burden from H358-FLuc/iDAb RAS-UBOX (**c**) and H358-FLuc/VHL-DP KRAS (**d**) was assessed by bioluminescence imaging at the end of the experiment (day 20). Photon flux (i.e. luminescence signal) was quantified for each group at day 20 (mean ± SEM) from $n = 4$ and $n = 5$ mice for the no dox and dox conditions respectively. **e, f** Western blot analysis of H358 tumour lysates from resected H358-FLuc/iDAb RAS-UBOX tumours (**e**) or from resected H358-FLuc/VHL-DP KRAS tumours (**f**) after 48 h of dox treatment compared to non-treated H358 mice tumours taken at the end point of the experiment (20 days). Experiments in (**e, f**) were performed once. Quantifications of pMEK/MEK and pERK/ERK signals normalised to condition in which dox was not added (No dox) are shown underneath individual lanes. The black arrowhead indicates the specific band corresponding to FLAG-tagged VHL-DP KRAS protein. **a–f** Source data are provided as a Source Data file.

The addition of warheads to intracellular antibodies such as fusing procaspase to induce apoptosis[54,55] or FBOX proteins to cause proteolysis[27,28] provides a mechanism by which macromolecules could be converted to potent macrodrugs. While PROTAC small molecules have been described and are KRAS[G12C] specific due to covalent interaction of compound to the protein[45], they did not degrade endogenous KRAS[G12C]. In addition, an affinity-directed protein missile system has shown degradation of KRAS and HRAS[56], but so far no isoform specific RAS degraders have been found.

In this study, we demonstrate a way forward to generalised KRAS inhibitors by engineering a KRAS-specific DARPin as a fusion protein with an E3 ligase to invoke proteasome targeting of KRAS and subsequent degradation. We have engineered our KRAS-specific DARPin with an E3 ligase warhead and compared this to an engineered pan-RAS binding iDAb[46]. We found that RAS degradation could readily be achieved with both macrodrugs but that the specific E3 ligase and the terminal location of the E3 ligase on the macrodrug was important. Interestingly, the VHL E3 ligase was most efficient with the KRAS binding DARPin and the UBOX domain from CHIP E3 ligase with the pan-RAS iDAb. Both degraders allowed the efficient depletion of endogenous RAS proteins in multiple cell lines (i.e. from lung, pancreas, bladder and connective tissue) suggesting a broad applicability of this strategy. Furthermore, we showed the high specificity of the KRAS degrader that only depletes KRAS without affecting HRAS or NRAS protein level in all the cell lines tested. Therefore, the properties of the RAS degraders would make them useful tools for studying RAS and KRAS biology in cells. It is possible that the degraders function by co-degradation with the target RAS protein or by a catalytic mechanism. Supplementary Fig. 5a shows that the protein levels of all iDAbs and DARPins (non-relevant and relevant) increased when a proteasome inhibitor was added suggesting that the turnover of these exogenous proteins is dependent on the proteasome machinery. In short time courses (up to 24/48 h, Fig. 2c), the effects of the degraders on RAS degradation could be catalytic as there was no apparent decrease of the degrader protein levels but at longer times (>48 h) the degrader protein levels decreased more than the negative controls, suggesting a co-degradation. As previously described, we also observed that the degrader technology can modify the potency and/or the selectivity of the parental binder[49–51] as presented in this study with the DP KRAS. We show that the parental DP KRAS affected RAS downstream signalling pathways and the proliferation of all the cell lines tested, showing no specificity towards mutant KRAS cells. This could be attributed to its ability to modulate mutant and wild-type KRAS on multiple levels[3]. However, once converted into a KRAS degrader, it only inhibited mutant KRAS cancer cells, while the pan-RAS degrader showed no specificity for any RAS isoform mutant protein like the parental binders iDAb RAS and DP KRAS.

An important finding of our study is that KRAS[WT] depletion by KRAS degrader did not lead to an inhibition of the proliferation of RAS[WT] cells, especially untransformed cells. These data are supported by a previous study showing that expression of only one RAS in RASless mouse embryonic fibroblast did not impede their ability to proliferate while only the removal of the three RAS isoforms stopped the growth of these engineered fibroblast cells[57]. This conclusion was confirmed in multiple cell lines with the pan-RAS degrader described here, strongly suggesting that mechanisms of compensation exist between the three RAS isoforms in RAS[WT] cells in a way that loss of expression of one (or two) of the isoforms can be overcome via the expression of the other isoform(s). In addition, our data highlighted the efficacy of our KRAS degrader in vivo, with a rapid regression of mutant KRAS tumours. Therefore, KRAS-targeted degradation is

an attractive therapeutic strategy for cancers with KRAS mutations, and not limited to any specific codon change.

Our study shows a proof of concept for the development of pan-KRAS specific degraders as therapeutics that could be implemented in any cancer with KRAS mutations. The direct therapeutic use of these inhibitors with an intracellular protein such as KRAS is currently limited, but efforts have been made for the delivery of macromolecules by mRNA[58] or other delivery strategies[59].

## Methods

**Cell culture**. A549, H358, HCT116, HEK293T, HT1080, MIA PaCa2 cells were grown in DMEM medium (Life Technologies), H1299, HCC827, T24 cells in RPMI medium (Life Technologies) and MRC5 in MEMα medium (Life Technologies). All cell lines were supplemented with 10% fetal bovine serum (FBS) (Sigma) and 1% penicillin/streptomycin (Life Technologies). Cells were grown at 37 °C with 5% $CO_2$. The cell lines were obtained from ATCC (catalogue numbers provided in the Reporting Summary) except MRC5, T24 and H1299, which were a gift from Prof. Geoff Higgins, University of Oxford, and MIA PaCa2 from Prof. Gillies McKenna. Characterisation of the cell lines was achieved by cloning and sequencing of the K, N and HRAS cDNA from each line. MIA PaCa2 was authenticated by short-tandem repeat DNA profiling services (ATTC). All cell lines were tested to confirm that they were free of mycoplasma.

**Cell transfection**. HEK293T cells were seeded in 6-well plates (650,000 cells per well) and 2 μg of plasmid DNA was transfected with Lipofectamine 2000 (Thermo-Fisher, see also "BRET2 methods" section below). HCT116 cells were seeded in 6-well plates (650,000 cells per well). Cells were transfected 24 h later with 2.5 μg of plasmid, 8.75 μL of Lipofectamine LTX and 2.5 μg of PLUS[TM] Reagent (Thermo-Fisher) for another 24 h before Western blot analysis.

**Molecular cloning**. All RAS cDNAs (KRAS[G12D], KRAS[WT], NRAS[Q61H] and HRAS[G12V]-CAAX) were cloned into the pEF-RLuc8-MCS, pEF-GFP²-MCS or pEF-3xFLAG-MCS plasmids, full-length CRAF[S257L] was cloned into pEF-GFP²-MCS and DARPin were cloned into the pEF-MCS-GFP² or pEF-MCS-mCherry plasmid. The cloning details of all these plasmids are described elsewhere[3,12,60].

Full-length VHL (amino acids 1–213) and UBOX domain (amino acids 128–303 from the CHIP E3 ligase) were cloned into PmlI/XhoI sites of the pEF-GFP²-MCS or into NotI/XbaI sites of the pEF-MCS-GFP² plasmids to replace the GFP² moiety. Linkers, DARPins and iDAbs were inserted into pEF-VHL-MCS and pEF-UBOX-MCS using XhoI/XbaI sites or into pEF-MCS-UBOX and pEF-MCS-VHL using PmlI/NotI sites. A single FLAG tag was added by polymerase chain reaction (PCR) at the carboxy terminal end of the DARPin degrader vectors or on the amino terminal end of the iDAb degrader vectors.

VHL-DP KRAS-FLAG, VHL-DP Ctl-FLAG, FLAG-iDAb RAS-UBOX and FLAG-iDAb Ctl-UBOX sequences were cloned in the TLCV2 lentivector (Addgene plasmid #87360[61]) by PCR using AgeI/NheI sites. Coding region DNA and protein sequences of these four constructs are shown in Supplementary Figs. 1e and 2–4.

**Lentivirus production**. For each virus produced, $4.5 \times 10^6$ HEK293T cells were seeded per 100 mm dish ($7 \times 100$ mm dishes per virus production) in 9 mL of complete DMEM. Twenty-four hours later, cells were transfected with 12 μg of the TLCV2 construct of interest (i.e. VHL-DP KRAS-FLAG, VHL-DP Ctl-FLAG, FLAG-iDAb RAS-UBOX and FLAG-iDAb Ctl-UBOX), 8 μg of psPAX2, 3 μg of pMD2.G (the latter are lentiviral packaging and envelope vectors, respectively) and 46 μL of Lipofectamine 2000 (quantities for one 100 mm dish). The supernatants were collected 48 h after transfection, centrifuged 5 min at $640 \times g$, filtered (0.45 μm filter) and centrifuged 2 h at $48,000 \times g$ at 4 °C. The virus from $7 \times 100$ mm dishes was resuspended in 250 μL of PBS.

**Viral transduction and macrodrug expression**. Cells were transduced with the appropriate lentivirus for 48 h in 6-well plate in 1 mL of medium containing 8 μg.mL⁻¹ of polybrene (Sigma, Cat#107689). Transduced cells were selected with puromycin (MP Biomedicals, Cat#194539). The puromycin concentrations used for selection of each cell line were 0.5 μg.mL⁻¹ for A549 and HCC827 cell lines, 0.75 μg.mL⁻¹ for H358, MIA PaCa2 and HT1080 cell lines, 0.8 μg.mL⁻¹ for H1299 and MRC5 cell lines and 1 μg.mL⁻¹ for T24 cell line.

Doxycycline (Sigma, Cat#D9891) was used to induce the expression of macrodrug E3-ligase or macrodrug GFP² fusions or controls from the TLCV2 lentivector. The doxycycline induction was carried out by addition of stock solution (100 μg.mL⁻¹) to culture medium and continued incubation at 37 °C. For proteasome inhibition, epoxomicin (Sigma, Cat#E3652) was used at 0.8 μM for 18 h followed by protein analysis.

**Establishment of H358 and H1299-FLuc stable clones**. H358 and H1299 cells expressing VHL-DP KRAS and iDAb RAS-UBOX were transfected with

pEF-FLuc[24] using Lipofectamine LTX following the manufacturer recommendations. After 48 h of transfection, cells were selected with 1 mg.mL$^{-1}$ of G418 (Sigma, Cat#A1720) and clones were picked and characterised.

**Quantitative real-time PCR (qRT-PCR)**. H358 cells were plated at $0.8 \times 10^6$ cells per well in a 6-well plate. After 24 h, the cells were treated or not with 0.2 µg.mL$^{-1}$ of doxycycline for 24 h. Cell were lysed in 1 mL of TRIzol reagent (Life Technologies) per 6 well. Total RNA was extracted with the Direct-zol$^{TM}$ RNA miniprep (Zymo Research) following the manufacturer's protocol. RNA was eluted with 15 µL of nuclease-free H$_2$O. cDNA was synthesised from 1.5 µg of total RNA per condition using SuperScript II Reverse Transcriptase (Invitrogen). RT-PCR was performed with 400 nM primers, diluted with 12.5 µL Fast SYBR Green Master Mix (Applied Biosystems) in a final volume of 25 µL. RT-PCR experiments were performed with the following protocol on a 7500 Fast (Applied Biosystems): 95 °C for 20 s, 40 cycles of 95 °C for 3 s, and 60 °C for 30 s. qRT-PCR samples were performed and analysed in duplicate, from two independent experiments. GAPDH was used for normalisation. Primers used in this study are as follows:

DUSP6For: 5′ CTCGGATCACTGGAGCCAAAAC 3′
DUSP6Rev: 5′ GTCACAGTGACTGAGCGGCTAA 3′
GAPDHFor: 5′ GTCTCCTCTGACTTCAACAGCG 3′
GAPDHRev: 5′ ACCACCCTGTTGCTGTAGCCAA 3′

**BRET2 assays and measurements**. For all BRET experiments (titration curves and competition assays) 650,000 HEK293T cells were seeded in each well of a 6-well plate. After 24 h at 37 °C, cells were transfected with a total of 1.6 µg of DNA mix (with donor + acceptor ± competitor plasmids), using Lipofectamine 2000 transfection reagent (Thermo-Fisher). Cells were detached after 24 h, washed with PBS and seeded in a white 96-well plate (clear bottom, PerkinElmer) in OptiMEM no phenol red medium complemented with 4% FBS. Cells were incubated for an additional 20–24 h at 37 °C before the BRET assay reading. A step-by-step protocol is provided elsewhere[60].

BRET2 signal was determined immediately after injection of coelenterazine 400a substrate (10 µM final) to cells (Cayman Chemicals), using a CLARIOstar instrument (BMG Labtech) with a luminescence module.

**2D and 3D cell proliferation assays**. Cells were seeded in white 96-well plates (clear bottom, PerkinElmer, Cat#6005181) for 2D-adherent proliferation assays or in ultra-low attachment 96-well plates (Corning, Cat#7007) for 3D spheroid assays. All cell seeding was optimised to maintain linear growth over the time of the assay. The following day, a 10× doxycycline solution was prepared (1–2 µg.mL$^{-1}$ for 0.1–0.2 µg.mL$^{-1}$ final concentration). Cells were incubated in the presence of the doxycycline for 6 days. Cell viability was analysed every two days using CellTiter-Glo (Promega, Cat#G7572) by incubation with the cells for 15 min. Cell viability was determined by normalising doxycycline-treated cells to non-treated cells. Cells from the ultra-low attachment plates were transferred into a white 96-well plate (Greiner, Cat#655075) before reading on a CLARIOstar instrument.

**Cell growth assay of H358-FLuc and H1299-FLuc clones**. Totally, 40,000 of H358-FLuc (iDAb RAS-UBOX or VHL-DP KRAS) or 50,000 of H1299-FLuc (iDAb RAS-UBOX or VHL-DP KRAS) cells were seeded per well of 6-well plate (each condition done in duplicate). After 24 h, medium alone (−dox) or medium containing 0.2 µg.mL$^{-1}$ doxycycline (+dox) was added in each well. Viable cells were counted with a haematocytometer and trypan blue every 2 days.

**Immuno-precipitation assay**. HEK293T cells were transfected for 24 h with pEF-3xFLAG-KRAS$^{WT}$ or pEF-3xFLAG-KRAS$^{G12D}$ and pEF-DARPins-GFP$^2$ plasmids. Cells were washed once with PBS and lysed in the immuno-precipitation buffer (150 mM NaCl, 50 mM Tris-HCl pH 7.4, 10 mM MgCl$_2$, 10% glycerol and 0.5% Triton X-100) supplemented with protease inhibitors (Sigma, Cat#P8340) and phosphatase inhibitors (Thermo-Fisher, Cat#1862495) for 20 min. Lysates were centrifuged for 15 min and the supernatant incubated with protein G magnetic beads (Life Technologies, Cat#10004D) and anti-FLAG antibody (Sigma, Cat#F3165). The complexes were incubated for 4 h at 4 °C with rotation. Beads were washed 5 times with the IP buffer, before the bound proteins were eluted with 1× loading buffer and resolved on 12.5% sodium dodecyl sulfate polyacrylamide gel electrophoresis (SDS-PAGE).

**Western blot analysis**. Cells were washed once with PBS and lysed in SDS-Tris buffer (STB: 1% SDS, 10 mM Tris-HCl pH 7.4) supplemented with protease inhibitors (Sigma) and phosphatase inhibitors (Thermo-Fisher). Cell lysates were sonicated with a Branson Sonifier.

Mouse tumours were lysed in the radioimmunoprecipitation assay buffer (150 mM NaCl, 1.0% Triton X-100, 0.5% sodium deoxycholate, 0.1% SDS, 50 mM Tris, pH 8.0) with a ratio of 200 µL of lysis buffer for 10 mg of tumour and homogenised with an electric disperser (T10 basic ULTRA-TURRAX, IKA) until the tissue was liquefied. The lysate was incubated on ice for 1 h, followed by centrifugation at $16,100 \times g$ at 4 °C and the supernatant was collected. The protein concentrations from cell and tumour

lysates were determined by using the Pierce BCA protein assay kit (Thermo-Fisher). Equal amounts of protein (20–50 µg) were resolved on 10 or 12.5% SDS-PAGE and subsequently transferred onto a polyvinylidene fluoride membrane (GE Healthcare). The membrane was blocked either with 10% non-fat milk (Sigma, Cat#70166) or 10% bovine serum albumin (Sigma, Cat#A9647) in TBS-0.1% Tween20 and incubated overnight with primary antibody at 4 °C. After washing, the membrane was incubated with horse radish peroxidase-conjugated secondary antibody for 1 h at 20 °C. The membrane was washed with TBS-0.1% Tween and developed using Clarity Western ECL Substrate (Bio-Rad) and CL-XPosure films (Thermo-Fisher) or the ChemiDoc XRS + imaging system (Bio-Rad).

Primary antibodies include anti-phospho-p44/22 MAPK (pERK1/2) (1/4000, CST, Cat#9101S), anti-p44/42 MAPK (total ERK1/2) (1/1000, CST, Cat#9102S), anti-phospho-MEK1/2 (1/2000, CST, Cat#9154S), anti-MEK1/2 (1/500, CST, Cat#4694S), anti-phospho-AKT S473 (1/1000, CST, Cat#4058S), anti-AKT (1/1000, CST, Cat#9272S), anti-pan-RAS (1/200, Millipore, Cat#OP40), anti-KRAS (1/100, Santa Cruz Biotechnologies, Cat#sc-30), anti-NRAS (1/100, Santa Cruz Biotechnologies, Cat#sc-31 and 1/3000, Abcam, Cat#ab77392), anti-HRAS (1/500, Proteintech, Cat#18295–1-AP), anti-cleaved PARP (1/1000, CST, Cat#9541), anti-cleaved caspase 3 (1/500, CST, Cat#9664), anti-GFP (1/500, Santa Cruz Biotechnologies, Cat#sc-9996), anti-FLAG (1/2000, Sigma, Cat#F3165), anti-β-actin (1/5000, Sigma, Cat#A1978) and anti-αtubulin (1/2000, Abcam, Cat#ab4074). Secondary antibodies include anti-mouse IgG HRP-linked (CST, Cat#7076), anti-rabbit IgG HRP-linked (CST, Cat#7074) and anti-goat IgG HRP-linked (Santa Cruz Biotechnologies, Cat#2354).

**Human tumour xenograft assay in nude mice**. Totally, $4.5 \times 10^6$ H358-FLuc expressing either iDAb RAS-UBOX clone B2 or VHL-DP KRAS clone F7 or $5 \times 10^6$ H1299-FLuc expressing either iDAb RAS-UBOX clone E1 or VHL-DP KRAS clone E5 were injected subcutaneously into the left flank of 5–7-week-old female CD-1 athymic nude mice (Charles River). The mice were fed with normal diet and water until their subcutaneous tumour reached 3–4 mm diameter (2–3 mm for H1299 cells), approximately 18 days after injection. The mice were divided into 2 groups of 5 mice (3 mice for H1299-FLuc/VHL-DP KRAS), one of which was supplied with dox (Sigma) via drinking water (2 mg.mL$^{-1}$ in 20% black-currant juice) and doxycycline diet (200 mg.kg$^{-1}$, Special Diets Services). Note on the first day of doxycycline treatment, mice were injected with 100 µL of 4 mg.mL$^{-1}$ of doxycycline in sterile 0.9% aqueous NaCl by intraperitoneal injection. One mouse of each of the no doxycycline control groups injected with H358-FLuc/VHL-DP KRAS, H358-FLuc/iDAb RAS-UBOX and H1299-FLuc/VHL-DP KRAS was excluded from analysis due to lack of tumour development. Subcutaneous tumour growth was monitored by measuring three times weekly with digital calipers (Thermo-Fisher) and by bioluminescence as described below. Tumour volume was calculated using the formula: tumour volume = $(L \times W^2)/2$, in which $L$ and $W$ refer to the length and width of the tumour, respectively. Animals were culled in accordance with Home Office licence restrictions. After humane sacrifice, the mice were dissected for tumours sampling. The University of Oxford Ethical Review Committee approved the study protocol described in the manuscript.

**In vivo bioluminescence imaging (BLI)**. Bioluminescence was measured in transplanted subcutaneous tumours non-invasively using the IVIS Lumina imaging system (PerkinElmer) following injection of the luciferase substrate D-luciferin. All of the images were taken after intraperitoneal injection of 150 µL of D-luciferin (stock solution at 30 mg.mL$^{-1}$ in DPBS, PerkinElmer) corresponding to 150 mg.kg$^{-1}$ body weight of D-luciferin. BLI were acquired after 5 min of D-luciferin injection. During image acquisition, mice were sedated continuously by inhalation of 3% isoflurane. Image analysis and bioluminescence quantification was performed using Living Image software (PerkinElmer).

**Quantification and statistical analysis**. Quantifications were performed using Image Lab (Biorad), Prism 8.0 (GraphPad Software) or Living Image (PerkinElmer). BRET titration curves and statistical analysis were performed using Prism 8.0 (GraphPad Software). The data are typically presented as mean ± SD or SEM as specified in the figure legends. Statistical analyses were performed with an unpaired two-tail Student's $t$ test. ns non-significant, $*P < 0.05$.

**Reporting summary**. Further information on research design is available in the Nature Research Reporting Summary linked to this article.

## Data availability

All relevant data are within the paper and its Supplementary Information file and in a Source data file. The source data underlying Figs. 1a–f and 2–8 and Supplementary Figs. 1b–d and 5–9 are provided as a Source Data file. Additional data supporting the conclusions are available from the corresponding author on reasonable request.

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

## Acknowledgements

Grant numbers Wellcome Trust Award 099246/Z/12/Z; Bloodwise grant 12051; MRC MR/J000612/1. We would like to thank Dr. Remco Prevo and Prof. Geoff Higgins for T24, H1299 and MRC5 cells. We acknowledge the expert technical help from Dr. Hedia Chagraoui for the qPCR experiments and analysis and Dr. Alex Martin and Prof. Robert Carlisle

for access to the IVIS instrument and help with its operation. We also like to thank Roo Bhasin and Jonathon Merrill for excellent technical help with the mouse experiments.

## Author contributions

Originators of project: N.B. and T.R. Participated in research design: N.B. and T.R. Conducted experiments: N.B. and A.M. Performed data analysis: N.B., A.M. and T.R. Wrote or contributed to the writing of the paper: all authors.

## Competing interests

The authors declare no competing interests.
