## [Peer Review File · Nature Communications]

Review of ***A potent KRAS macromolecule degrader specifically targeting tumours with mutant KRAS***

Many tumours are able to survive and grow due to overactivation signalling of the MAP kinase pathway which causes an overexpression of genes responsible for cell proliferation and survival. This pathway is regulated by KRAS, which is frequently mutated in tumours, leading to elevated level of active KRAS and uncontrolled downstream signalling. Therapeutic targeting of KRAS therefore remains one of the holy grails of cancer treatment. Patients bearing mutations on KRAS represent high unmet medical need, yet KRAS remains a so-called undruggable target and only recently small molecules targeting KRAS^{G12C} covalently have entered the clinic, though might be limited by pathway adaptation (Xue et al. Nature 2020). Targeting KRAS for degradation may thus represent an alternative attractive therapeutic concept.

Here the authors have targeted RAS proteins for intracellular degradation by stably expressing fusion constructs of ankyrin-based KRAS-binding antibodies and the E3 ligase VHL, and comparing these with a pan-RAS targeting single-domain antibody fused to a UBOX domain of the E3 ligase CHIP. After showing the constructs work as anticipated and identifying the most active ones, they show both in cell lines and in vivo that the KRAS-specific degrader constructs results in durable reduction of pERK and other phosphor-kinase signalling pathways, and consequently induction of apoptosis and suppression of proliferation in KRAS-dependent mutant cell lines only. That is however not found to be the case in cell lines expressing RAS WT and other RAS isoforms mutants, despite the construct inducing complete degradation of all KRAS forms (i.e. mutants and wild-type).

In contrast, the UBOX-CHIP fusion construct, which is a pan-RAS degrader, affects the downstream pathway in a wider range of cell lines, including WT RAS. The authors thus conclude that upon selective degradation of KRAS, growth is inhibited only in KRAS-mutant driven cancer cells.

The data provides strong support for an important claim - that KRAS selective degradation may provide an impactful strategy to address KRAS dependent mutant tumours, in a safe and non-toxic manner by sparing non- mutant or RAS independent cells. While the technology of fusing binding domains such as DARPINs to E3 ligases is not novel, here it is applied in an elegant and innovative fashion to make an important advance. The methods are sound and the experiments are biologically relevant, logical and performed and analysed well, also using proper controls (proteasomal inhibition, monitoring FLAG for expression of the construct, designing parental construct as binding but non-degrading controls and so on). The results are presented in a clear and logical way.

In our opinion, the advance reported has the significance, validity and timeliness to warrant publication and so are supportive of publication in Nature Communication.

We would like to offer the authors some comments for consideration to strengthen their paper.

- Referencing is good throughout, but could improve. The authors duly cite relevant and important RAS literature, including on the G12C targeting inhibitors. They might want to include references to the most recent articles describing the most advance compounds from Mirati and Amgen (<https://doi.org/10.1021/acs.jmedchem.9b02052> and <https://doi.org/10.1021/acs.jmedchem.9b01180>). As the papers advocate on pan-KRAS targeting, the authors might want to mention recent literature describing a pan KRAS inhibitor (<https://doi.org/10.1073/pnas.1904529116>). Moreover, references 29-34 are all

from the same group and could benefit here with more balanced citations of other Labs contributors in the PROTAC field.

- Protein levels are nicely quantified and plotted in Figure 1 and 5i, but are not quantified in the remaining figures. The authors might want to consider quantifying protein levels also in the other figures, as in some cases the effect of the degradation of pERK etc suppression was feeble. this could be easily done by reporting values as numbers underneath the blot images
- Fig S2a - protein level of IDAB/DP seems to go up in the presence of proteasome inhibitor - the authors might want to comment on what that might be?
- pg. 8 and SI Fig 3 and SI Fig 4: unclear the rationale/motivation for studying the effect of the GFP-fusion - presumably this is to control for binding to Res without inducing its degradation? if so, this should be made clearer in the text. Then why is RAS degradation induced in some cases using these binding controls e.g. Fig. S3a (H358), c (A549) and f (T24)?
- can the authors provide an explanation as to why the parental KRAS binder (DARPin KRAS-GFP) decreased the proliferation of all the stable cell lines and not only the ones with KRAS mutation?
- The authors show that A549 proliferation is only blocked by the pan-RAS degrader and not as much by the KRAS-selective degrader, and similarly that apoptosis is observed only with the pan but not with the KRAS-selective degrader in this cell line. They then (on pg. 10) define A549 as "KRAS independent cells". Is this definition based on their own results, or is this known in literature? in the latter case they should give citation(s)
- it might have been interesting to perform the in vivo experiments with the control KRAS-GFP binders to assess the extent to which induced degradation fares compared to blockade. Please comment
- source and authentication of all cell lines should be provided

other technical notes:

- Figure 2. panels e and f are missing a loading control
- In figure 3b authors are using separate antibodies for K/N/H RAS to figure out what protein is being degraded, whereas in figure 4. and 5. they use panRAS antibody. In the figure 6. they switch back to KRAS antibody and in the animal experiments they are back to panRAS antibodies. This can be misleading and not ideal, since we don't know if K/N/H proteins are expressed equally in the cell. Perhaps the authors might like to comment as to why separate K/N/H antibodies are not used throughout the manuscript.
- authors show cleaved caspase 3 and PARP as a proof of activating cell death, but this would be clearer if full caspase 3 and PARP would be shown, especially in cases of control cell lines.
- bottom of pg. 15: full-length VHL should specify this is the long iso form i.e. 1-213

We agree to waive our anonymity,
Alessio Ciulli and Vesna Vetma

Reviewer #2 (Remarks to the Author):

Overall, this is a very interesting and timely paper on targeted degradation of RAS proteins. While the system used is a model and not directly therapeutically exploitable, it is conceptually very important because it establishes the feasibility of KRAS isoform selective degradation and shows that this has far fewer effects on wild type cells than does a pan RAS degrader. The paper establishes the potential of KRAS isoform selective targeted degraders as future therapies for KRAS mutant cancers.

Some specific comments:

1. Is it possible to determine whether it is the RAS proteins that become ubiquitinated, rather than the complexed DARPin or the iDAb? In previous work on KRAS targeted degraders (<https://doi.org/10.1016/j.chembiol.2019.12.006>) there was a problem with failure to cause KRAS ubiquitination, but rather getting ubiquitination of the GFP tag. Given this, if KRAS is being ubiquitinated, what lysine is being targeted?
2. Do the DARPin or the iDAb become ubiquitinated and degraded during the process of RAS degradation – are the effects of the degraders single turnover or catalytic?
3. (minor point) In Supplementary figure 1 (e-h), the sequences could be mapped out more clearly. It would be good to show where the VHL and the UBOX sequences start and end, also where is the KRAS DARPin and the iDAb-RAS and the linker involved.

Review of A potent KRAS macromolecule degrader specifically targeting tumours with mutant KRAS

Reviewer #1

Many tumours are able to survive and grow due to overactivation signalling of the MAP kinase pathway which causes an overexpression of genes responsible for cell proliferation and survival. This pathway is regulated by KRAS, which is frequently mutated in tumours, leading to elevated level of active KRAS and uncontrolled downstream signalling. Therapeutic targeting of KRAS therefore remains one of the holy grails of cancer treatment. Patients bearing mutations on KRAS represent high unmet medical need, yet KRAS remains a so-called undruggable target and only recently small molecules targeting KRASG12C covalently have entered the clinic, though might be limited by pathway adaptation (Xue et al. Nature 2020). Targeting KRAS for degradation may thus represent an alternative attractive therapeutic concept.

Here the authors have targeted RAS proteins for intracellular degradation by stably expressing fusion constructs of ankyrin-based KRAS-binding antibodies and the E3 ligase VHL, and comparing these with a pan-RAS targeting single-domain antibody fused to a UBOX domain of the E3 ligase CHIP.

After showing the constructs work as anticipated and identifying the most active ones, they show both in cell lines and in vivo that the KRAS-specific degrader constructs results in durable reduction of pERK and other phosphor-kinase signalling pathways, and consequently induction of apoptosis and suppression of proliferation in KRAS-dependent mutant cell lines only. That is however not found to be the case in cell lines expressing RAS WT and other RAS isoforms mutants, despite the construct inducing complete degradation of all KRAS forms (i.e. mutants and wild-type).

In contrast, the UBOX-CHIP fusion construct, which is a pan-RAS degrader, affects the downstream pathway in a wider range of cell lines, including WT RAS. The authors thus conclude that upon selective degradation of KRAS, growth is inhibited only in KRAS-mutant driven cancer cells. The data provides strong support for an important claim - that KRAS selective degradation may provide an impactful strategy to address KRAS dependent mutant tumours, in a safe and non-toxic manner by sparing non-mutant or RAS independent cells. While the technology of fusing binding domains such as DARPINs to E3 ligases is not novel, here it is applied in an elegant and innovative fashion to make an important advance. The methods are sound and the experiments are biologically relevant, logical and performed and analysed well, also using proper controls (proteasomal inhibition, monitoring FLAG for expression of the construct, designing parental construct as binding but non-degrading controls and so on). The results are presented in a clear and logical way.

In our opinion, the advance reported has the significance, validity and timeliness to warrant publication and so are supportive of publication in Nature Communication.

We would like to offer the authors some comments for consideration to strengthen their paper.

- Referencing is good throughout, but could improve. The authors duly cite relevant and important RAS literature, including on the G12C targeting inhibitors. They might want to include references to the most recent articles describing the most advance compounds from Mirati and Amgen (<https://doi.org/10.1021/acs.jmedchem.9b02052> and <https://doi.org/10.1021/acs.jmedchem.9b01180>).

These references have been added in the introduction of the manuscript as references 20 and 21.

As the papers advocate on pan-KRAS targeting, the authors might want to mention recent literature describing a pan KRAS inhibitor (<https://doi.org/10.1073/pnas.1904529116>).

This reference has been added in the manuscript, reference 13.

Moreover, references 29-34 are all from the same group and could benefit here with more balanced citations of other Labs contributors in the PROTAC field.

We added references from different groups working on the PROTAC field with references 37-39 (doi:10.1016/j.chembiol.2019.10.013; doi:10.1016/j.chembiol.2019.11.014; doi:10.1038/s41467-019-11429-w) and references 42-43 (doi:10.1021/acscchembio.9b00505; doi:10.1038/s41589-019-0294-6).

- Protein levels are nicely quantified and plotted in Figure 1 and 5i, but are not quantified in the remaining figures. The authors might want to consider quantifying protein levels also in the other figures, as in some cases the effect of the degradation of pERK etc suppression was feeble. this could be easily done by reporting values as numbers underneath the blot images.

Because Figure 4 is quantified in Figure 5i, we now provide the quantification for Figures 2 & 8. We had not done quantification of Figure 3 because the data with loss of RAS signal are clear that we believe quantification is not needed.

- Fig S2a - protein level of IDAB/DP seems to go up in the presence of proteasome inhibitor - the authors might want to comment on what that might be?

The increase of iDAB/DP protein levels (both non-relevant and relevant ones) suggests that the turnover of these exogenous proteins could be dependent on the proteasome machinery.

- pg. 8 and SI Flg 3 and SI Fig 4: unclear the rationale/motivation for studying the effect of the GFP-fusion - presumably this is to control for binding to Res without inducing its degradation? if so, this should be made clearer in the text. Then why is RAS degradation induced in some cases using these binding controls e.g. Fig. S3a (H358), c (A549) and f (T24)?

The GFP fusions are a control in this study as it was shown in previous work that the conversion of a parental binder into a degrader can modify its specificity (doi: 10.1016/j.chembiol.2017.09.010; doi: 10.1038/s41589-018-0055-y; doi: 10.1038/nchembio.2538). We have added a sentence in the text to explain this on page 8. Furthermore, in the case of the DARPIn KRAS, the GFP fusion data further show the requirement for building a KRAS-specific degrader rather than simply using the KRAS parental binder (please also see our reply on the next point below).

Only the pan-RAS binder induces a decrease of RAS level in some cell lines. It seems to be a cell line dependent observation, where some cells can be sensitive to the level of expression of the iDAB RAS. Furthermore, we previously showed that overexpression of iDAB RAS can decrease exogenous RAS protein level in HEK293T cells in some conditions (doi: 10.7554/eLife.37122).

- can the authors provide an explanation as to why the parental KRAS binder (DARPIn KRASGFP) decreased the proliferation of all the stable cell lines and not only the ones with KRAS mutation?

The KRAS binder binds to both mutant and wild-type KRAS. We previously showed that this KRAS-specific DARPIn binds to an interface remote from the switch regions (doi: 10.1038/s41467-019-10419-2) and interferes with KRAS functions by inhibiting the binding of GAP and GEF proteins and the dimerization of KRAS. All these inhibitory mechanisms can affect differently KRAS^{WT} and deregulate RAS downstream signalling (as shown in Supplementary Figure 4) and will consequently modulate the cell proliferation. These data further highlight the interest of making a KRAS specific degrader rather than simply using this KRAS parental binder. We added an explanation in the discussion page 13.

- The authors show that A549 proliferation is only blocked by the pan-RAS degrader and not as much by the KRAS-selective degrader, and similarly that apoptosis is observed only with the pan but not with the KRAS-selective degrader in this cell line. They then (on pg. 10) define A549 as “KRAS independent

cells". Is this definition based on their own results, or is this known in literature? in the latter case they should give citation(s).

This is based on previous studies. We added the corresponding references 52, 53 (doi:10.1016/j.ccr.2009.03.022; doi:10.1038/s41467-018-07644-6).

- it might have been interesting to perform the *in vivo* experiments with the control KRAS-GFP binders to assess the extent to which induced degradation fares compared to blockade. Please comment.

This would have been an interesting additional control in our *in vivo* study but we were not able to add extra conditions in our experiment due to technical limitations (although eight different conditions were used already).

- source and authentication of all cell lines should be provided.

The cell lines were obtained from ATCC and except MRC5, T24 and H1299 that were a gift from Prof. Geoff Higgins, University of Oxford and MiaPaca2 from Prof Gillies McKenna. The status of K, H and NRAS mutations in all the cell was confirmed by RT-PCR, cloning of the products and DNA sequencing. This statement has been added to the methods.

Other technical notes:

- Figure 2. panels e and f are missing a loading control.

The loading controls in panels e and f are the same as those in panels c and d because we split them for clarity. We have amended the figure legend to make this point clearer.

- In figure 3b authors are using separate antibodies for K/N/H RAS to figure out what protein is being degraded, whereas in figure 4. and 5. they use panRAS antibody. In the figure 6. They switch back to KRAS antibody and in the animal experiments they are back to panRAS antibodies. This can be misleading and not ideal, since we don't know if K/N/H proteins are expressed equally in the cell. Perhaps the authors might like to comment as to why separate K/N/H antibodies are not used throughout the manuscript.

The degradation induced by the pan-RAS and KRAS-specific degraders has been shown with specific KRAS/NRAS/HRAS antibodies in all the cell lines tested in this study (Figure 1, Figure 3 and Figure S6A). In the subsequent figures, it was used as a simple control (especially Figures 4 & 5 that show experiments performed with the same experimental conditions as in Figure 3) or we were only able to use one antibody because of technical issues (e.g. a limited quantity of materials).

- authors show cleaved caspase 3 and PARP as a proof of activating cell death, but this would be clearer if full caspase 3 and PARP would be shown, especially in cases of control cell lines.

In our study, we used cleaved caspase 3 and PARP antibodies to show a qualitative induction of apoptosis by the degraders in different cell lines rather than quantifying the apoptosis induction between different conditions. In addition to the loading control, we observe the background binding of antibody to either full cPARP or cCASP3 in most of the lanes. Therefore, we did not perform the total caspase 3 and PARP antibodies. We could include longer ECL exposures of the Western blots as a new Supplementary Figure if required.

- bottom of pg. 15: full-length VHL should specify this is the long iso form i.e. 1-213.

Yes, it is the long isoform (1-213), we added this information in the methods section p16.

We agree to waive our anonymity,
Alessio Ciulli and Vesna Vetma

Reviewer #2 (Remarks to the Author):

Overall, this is a very interesting and timely paper on targeted degradation of RAS proteins. While the system used is a model and not directly therapeutically exploitable, it is conceptually very important because it establishes the feasibility of KRAS isoform selective degradation and shows that this has far fewer effects on wild type cells than does a pan RAS degrader. The paper establishes the potential of KRAS isoform selective targeted degraders as future therapies for KRAS mutant cancers.

Some specific comments:

1. Is it possible to determine whether it is the RAS proteins that become ubiquitinated, rather than the complexed DARPin or the iDAb? In previous work on KRAS targeted degraders (<https://doi.org/10.1016/j.chembiol.2019.12.006>) there was a problem with failure to cause KRAS ubiquitination, but rather getting ubiquitination of the GFP tag.

In our study, we tested two E3 ligases fused to iDAb and DARPin at the N or C terminal end and not all of the combination induces endogenous untagged RAS degradation (Figure 1), suggesting a specific degradation of the target rather than the complexed DARPin/iDAb. However, we cannot exclude the possibility that the iDAb/DARPin could also be ubiquitinated (see point 2 below).

Given this, if KRAS is being ubiquitinated, what lysine is being targeted?

Several lysine residues of KRAS have been shown to be ubiquitinated (doi: 10.1074/jbc.REV119.011068; doi: 10.1038/s41418-019-0395-5; doi: 10.1074/jbc.M113.531178) and more than one could be ubiquitinated. The software UbiProber indicates that lysine 5, 104, 128 and 147 could be ubiquitinated. However, this information is not crucial for our study as our point is to degrade RAS/KRAS and observe the cellular consequences of such degradations.

2. Do the DARPin or the iDAb become ubiquitinated and degraded during the process of RAS degradation – are the effects of the degraders single turnover or catalytic?

We cannot distinguish these two mechanisms. Supplementary Figure 2a shows a decrease in all iDAbs and DARPins (non-relevant and relevant) protein levels on proteasome inhibitor treatment showing that the turnover of these exogenous proteins is dependent on the proteasome machinery. Figure 2c, d suggests that in short time courses (up to 24/48 hours), the effect of the degraders on RAS degradation is catalytic (no apparent decrease of the degraders protein level) but at longer times (>48 hours) the degrader protein levels decreased more than the negative degraders, suggesting a co-degradation. As it is difficult to differentiate between the normal turnover and the co-degradation of the complex, the effects of the degraders could be both catalytic and single turnover.

We have added a short discussion on this on page p13.

3. (minor point) In Supplementary figure 1 (e-h), the sequences could be mapped out more clearly. It would be good to show where the VHL and the UBOX sequences start and end, also where is the KRAS DARPin and the iDAb-RAS and the linker involved.

Supplementary figure 1 has been amended accordingly.